# Targeting the Crosstalk of Immune Response and Vascular Smooth Muscle Cells Phenotype Switch for Arteriovenous Fistula Maturation

**DOI:** 10.3390/ijms231912012

**Published:** 2022-10-10

**Authors:** Vikrant Rai, Harbinder Singh, Devendra K. Agrawal

**Affiliations:** Department of Translational Research, Western University of Health Sciences, Pomona, CA 91766, USA

**Keywords:** AVF maturation failure, chronic inflammation, immune cell infiltration, pro-inflammatory cytokines, resolvins, VSMCs phenotypes

## Abstract

Plaque formation, thrombosis, and embolism are the underlying causes of acute cardiovascular events such as myocardial infarction and stroke while early thrombosis and stenosis are common pathologies for the maturation failure of arteriovenous fistula (AVF). Chronic inflammation is a common underlying pathogenesis mediated by innate and adaptive immune response involving infiltration of immune cells and secretion of pro- and anti-inflammatory cytokines. Impaired immune cell infiltration and change in vascular smooth muscle cell (VSMC) phenotype play a crucial role in the underlying pathophysiology. However, the change in the phenotype of VSMCs in a microenvironment of immune cell infiltration and increased secretion of cytokines have not been investigated. Since change in VSMC phenotype regulates vessel remodeling after intimal injury, in this study, we investigated the effect of macrophages and pro-inflammatory cytokines, IL-6, IL-1β, and TNF-α, on the change in VSMC phenotype under in vitro conditions. We also investigated the expression of the markers of VSMC phenotypes in arteries with atherosclerotic plaques and VSMCs isolated from control arteries. We found that the inhibition of cytokine downstream signaling may mitigate the effect of cytokines on the change in VSMCs phenotype. The results of this study support that regulating or targeting immune cell infiltration and function might be a therapeutic strategy to mitigate the effects of chronic inflammation to attenuate plaque formation, early thrombosis, and stenosis, and thus enhance AVF maturation.

## 1. Introduction

Atherosclerosis is a chronic inflammatory and debilitating disease involving the buildup of the plaque and arterial lumen obstruction precipitating acute events such as myocardial infarction and stroke. The pathogenesis of atherosclerosis is multifactorial involving the deposition of lipids in the intima, foam cell formation, and mediators of inflammation mediating plaque formation with narrowing and hardening of the arteries. Chronic inflammation within the plaque and apoptosis of vascular smooth muscle cells (VSMCs) render it vulnerable to rupture and the rupture of the plaque due to a weak fibrous cap precipitates thrombus and emboli formation [1,2,3,4,5,6,7,8]. Along with these, VSMCs also play an important role in plaque formation and vulnerability. The aberrant proliferation of VSMCs promotes plaque formation but the density of VSMCs decreases in vulnerable plaque causing weakening of fibrous plaque and plaque rupture, and thus the importance of VSMCs density in plaque stability. However, emerging studies reported that the VSMCs population is not homogenous in plaque and a phenotypic switch of VSMCs occurs during plaque development. Chronic inflammation in the plaque is mediated not only by immune cells, primarily macrophages, but also by macrophage-like VSMCs, altogether mediating the progression of atherosclerosis and plaque rupture [9,10,11,12,13].

The altered phenotypes of VSMCs show less or no specificity to VSMC markers and might result in three different VSMC phenotypes: (i) contractile VSMCs (actin-binding protein (SM22α/tagln)+, α-smooth muscle actin (α-SMA)+, smooth muscle cell myosin (SM-MHC/myh11)+, smoothelin+, and H-caldesmon+), (ii) synthetic/proliferative VSMCs (cytokeratin-8 and 18 (CK-8, CK-18)+, S100A4+, calmodulin+, ICAM-1+, and VCAM-1+), and (iii) dedifferentiated VSMCs secreting proinflammatory cytokines (CD11b+, CD68+, Kruppel-like factor 4 (KLF4)+). The potential underlying regulatory mechanisms have been discussed in the literature [9,10,11,12,13]. In addition to these phenotypes, foam cell-like, macrophage-like, myofibroblast-like, osteochondrogenic-like, and mesenchymal stem cell-like phenotypes have also been reported [14]. Further, during atherosclerosis progression, the interaction between VSMCs, endothelial cells (ECs), and immune cells and the effect of VSMCs plasticity on inflammation, plaque formation, and rupture have been evaluated [15,16,17]. Most of the studies have reported the role of KLF4 and the interaction of immune cells with VSMCs; however, studies investigating the effect of immune cells (macrophages) and cytokines on VSMCs phenotype in atherosclerosis are limited [18,19,20]. Thus, an in-depth investigation of the effect on immune cells, mainly macrophages, and cytokines secreted by these cells on VSMCs phenotype is warranted.

Macrophages play a critical role in the pathogenesis of atherosclerosis and are the major immune cells in an atherosclerotic plaque [5,21]. We recently reported that macrophages, cytokines, and chemokines participate in perivascular cuffing mediating inflammation, early thrombosis, and stenosis of the vessels leading to early failure of arteriovenous fistula (AVF) [22,23]. Further, increased expression of IL-6 and TNF-α in association with increased α-SMA in thrombosed AVF suggests a correlation between VSMCs proliferation and these atherogenic cytokines [24]. IL-1, IL-6, and TNF-α are the major proinflammatory cytokines released by M1 macrophages [25]. This suggests that the proinflammatory cytokines released by M1 macrophages may affect the proliferation and phenotype of VSMCs. In this study, we investigated the effects of proinflammatory cytokines (IL-6, IL-1β, and TNF-α) and of macrophages on VSMCs isolated from Yucatan microswine and Yucatan miniswine carotid and femoral arteries, with and without atherosclerotic plaques, using in vitro and co-culture experiments. These findings are important in the context of arteriovenous fistula (AVF) formation where excessive neointimal hyperplasia, chronic inflammation, and early thrombosis in association with immune cells and cytokines play a critical role in early AVF maturation failure [22,23,26,27]. The results of this study provide insight to delineate novel therapeutic targets for enhancing AVF maturation.

## 2. Results

### 2.1. Differential Expression of VSMCs Phenotype Markers and Cytokines Expression in Arteries with Plaque and Control Arteries

qRT-PCR analysis of arteries without plaque (control) and arteries with plaque for VSMC phenotype markers showed significantly increased mRNA expression of α-SMA (*p* = 0.0052), transgelin (*p* = 0.0067) (markers of contractile VSMCs), cytokeratin-18 (*p* = 0.009), S100A4 (*p* = 0.00019), calmodulin 2 (*p* = 0.0019), ICAM-1 (*p* = 0.0011) (markers of synthetic/proliferative VSMCs), and KLF-4 (*p* = 0.0026) (marker for de-differentiated VSMCs) in plaque arteries compared to control. There was no significant difference in the mRNA expression of contractile VSMCs marker Myh11 (*p* = 0.288) (Figure 1).

RT-qPCR analysis also revealed increased mRNA expression of VCAM1 (*p* = 0.00013), CD11b (*p* = 0.003), and CD68 (*p* = 0.00091) in arteries with plaque compared to control arteries (Figure 2A). Additionally, the mRNA expression for TNF-α (*p* = 0.002), IL-6 (*p* = 0.0001), IL-8 (*p* = 0.008), and MCP1 (*p* = 0.0001) was also significantly increased in arteries with plaque compared to arteries without plaque (Figure 2B).

### 2.2. RT-qPCR for mRNA Expression of VSMCs Phenotypes Revealed Differential Expression in VSMCs Isolated from Arteries with and without Plaque

RT-qPCR studies revealed a decreased expression for α-SMA, increased expression for transgelin, significantly decreased expression of Myh11 (*p* = 0.006) and cytokeratin (CK)-18 (*p* = 0.008), significantly increased expression of S100A4 (*p* = 0.0001), significantly decreased expression of calmodulin (CALM) 2 (*p* = 0.002), decreased expression of ICAM1 and increased expression of KLF4. The changes in the mRNA expression of α-SMA, transgelin, ICAM1, and KLF4 were not significant (Figure 3). Furthermore, arteries with plaques showed significantly decreased expression of VCAM1, CD11b, and CD68 compared to control VSMCs (Appendix A).

### 2.3. Differential Response of Vascular Smooth Muscles at the Gene Level to Pro-Inflammatory Cytokines

To investigate the effect of pro-inflammatory cytokines, IL-6, IL-1β, and TNF-α, (most common atherogenic cytokines secreted by immune cells, mainly macrophages) on VSMCs phenotype, we treated VSMCs with different doses of cytokines and the cytokine dose with the best effect was chosen for further experiments. The cytokine dose titration for IL-6, IL-1β, and TNF-α showed different effects on VSMCs phenotype markers (Appendix A). Based on the results, doses of IL-6 (10 ng/mL), IL-1β (100 ng/mL), and TNF-α (100 ng/mL) were used for further experiments. The RT-qPCR analysis of the control VSMCs treated with IL-6, IL-1β, and TNF-α showed differential expression of VSMCs phenotype markers. The mRNA expression of α-SMA was increased with TNF-α and significantly increased with IL-6 and IL-1β while the expression of transgelin was decreased with TNF-α and IL-1β but increased with IL-6 though not significantly. Myh11 mRNA expression was increased with IL-6, IL-1β, and TNF-α but significantly only with IL-1β (Figure 4A). Similarly, the mRNA expression of CK-18 was significantly increased by IL-6, IL-1β, and TNF-α; of S100A4 by IL-6; of CALM2 by IL-6, IL-1β, and TNF-α; of KLF4 by IL-6; and CD68 by TNF-α. The effects of IL-6, IL-1β, and TNF-α varied with doses and for each marker (Figure 4B,C).

### 2.4. Co-Culture of VSMCs with M1 Macrophages Significantly Increased mRNA Expression of VSMCs Phenotype Markers

RT-qPCR analysis of the cDNA prepared from VSMCs co-cultured with M1 macrophages for 24 h revealed significantly increased expression of α-SMA, transgelin, Myh11, CK-18, S100A4, calmodulin 2 (CALM2), ICAM1, and KLF4 (Figure 5).

### 2.5. Pro-Inflammatory Cytokines, IL-6, IL-1β, and TNF-α, Regulate the Protein Expression of VSMC Phenotype Markers

VSMCs treated with proinflammatory cytokines IL-6, IL-1β, and TNF-α showed modulatory effects on VSMCs phenotype (Figure 6). The Western blot analysis [28] showed the highest increase in S100A4, a marker for proliferative phenotype, with IL-6 and TNF-α. The pro-inflammatory cytokines IL-6 and TNF-α also increased the protein expression of CK-18 and KLF4, the markers for proliferative and dedifferentiated (functioning like immune cells) phenotype. TNF-α attenuated the expression of α-SMA and IL-1β attenuated the protein expression of KLF-4 (Figure 6).

### 2.6. Inhibition of the Cytoplasmic Kinases Attenuated the mRNA Expression of VSMCs Phenotype Markers Even in the Presence of Pro-Inflammatory Cytokines

With the concept that pro-inflammatory cytokines signal through cytoplasmic kinases, we inhibited cytoplasmic kinases using LY294002 (PI3K inhibitor), SP600125 (JNK inhibitor), BAY 118,075 (NF-κB inhibitor), PDTC (NF-κB inhibitor), and UO-126 (ERK inhibitor) followed by treatment with IL-6, IL-1β, and TNF-α and checked the change in mRNA expression of VSMCs phenotype markers including α-SMA, transgelin, Myh11, CK-18, S100A4, calmodulin 2 (CALM2), ICAM1, VACM1, and KLF4. Stimulation with cytokines and inhibition of cytoplasmic kinases showed significantly decreased mRNA expression for VSMCs phenotype markers even in the presence of IL-6 (Figure 7), IL-1β (Figure 8), and TNF-α (Figure 9). These data suggest that inhibition of downstream signaling of pro-inflammatory cytokines mitigates or nullifies the effects of pro-inflammatory cytokines on the change in VSMC phenotypes.

### 2.7. Dual-Immunofluorescence Staining Showed Colocalization of α-SMA with Immune Cells and Pro-Inflammatory Cytokines Staining

Our RT-qPCR results revealed differential expression of VSMCs phenotype markers in control and arteries with plaque. Further, results suggest the effects of pro-inflammatory cytokines and the effect of inhibiting downstream signaling on VSMCs phenotype markers. To investigate whether VSMCs acquired different phenotypes, mainly de-differentiated VMCSs perpetuating inflammation, we dual-stained the control and plaque arteries with α-SMA with CD68, pro-inflammatory cytokines, and CD68 with pro-inflammatory cytokines. Dual-immunofluorescence showed higher positivity for α-SMA and CD68 in arteries with plaques as well as dual positivity (co-localized) of some VSMCs with CD68 positive staining in arteries with plaque (Figure 10).

Dual-immunofluorescence staining of arteries with plaque for α-SMA and pro-inflammatory cytokines showed co-localized positivity for both α-SMA and pro-inflammatory cytokines (Figure 11).

Additionally, dual-immunofluorescence of control and plaque arteries showed higher immunopositivity for CD68, IL-6, IL-1β, TNF-α, and IL-2 in arteries with plaque compared to control arteries. Further, co-localized staining for CD68 with IL-6, IL-1β, TNF-α, and IL-2 was also noted; however, all CD68 positive cells were not positive for IL-6, IL-1β, TNF-α, and IL-2 (Figure 12).

## 3. Discussion

A heterogeneous population of innate and adaptive immune cells plays an important role in regulating maturation failure of arteriovenous fistula (AVF) [26,27]. The role of dendritic cells, macrophages, and T cells in plaque formation and early thrombosis in AVF have been documented [4,5,26,27]. Infiltrating immune cells secrete pro- and anti-inflammatory cytokines regulating the inflammatory and resolution phase of healing. Impairment of this regulation results in chronic inflammation mediating plaque formation, early thrombosis, stenosis resulting in acute cardiovascular events, and AVF maturation failure. VSMCs play a critical role in these events and a switch in VSMC phenotype may affect these molecular processes to either enhance the favorable mechanism or impede the process [8,11,16,17]. There are reports in the literature on the role of VSMCs plasticity and phenotype switch in regulating plaque progression, atherosclerosis progression, vessel wall inflammation, adverse remodeling, and arterial calcification and targeting VSMCs plasticity and phenotype switch [16,19,29,30,31,32]. However, the role of VSMCs in vascular remodeling in AVF maturation has not been reported.

We previously reported the critical role of innate and adaptive immune response with a heterogenous population of immune cells and inflammation in early thrombosis of AVF fistula causing maturation failure [23,26,27]. In this study, we investigated the effect of pro-inflammatory cytokines on change in VSMCs phenotype. The results suggest that IL-6, IL-1β, and TNF-α have differential effects on VSMCs phenotype markers. A significantly increased expression of markers of proliferative (αSMA, CK-18, S100A4, CALM2, ICAM1, and VCAM1) and de-differentiated (KLF4) VSMCs [9,10,13] in arteries with plaque suggest the presence of ongoing process of vessel remodeling after intimal injury due to AVF creation and increased shear stress. A decreased mRNA expression of Myh11 and TAGLN suggests fewer contractile VSMCs. Furthermore, significantly increased mRNA expression of KLF4 indicates a higher number of de-differentiated VSMC. Since de-differentiated VSMCs mimic monocyte/macrophages like immune cells [16], it is reasonable to speculate that chronic inflammation causing early thrombosis in AVF may also be caused, in part, by these VSMCs along with infiltrating immune cells [22,23,26,27].

Chronic inflammation in arteries with thrombosis and stenosis is suggested by increased expression of CD68, CD11b, IL-6, IL-1β, and TNF-α. Significantly increased mRNA expression of CD11b and CD68 may be due to infiltrating immune cells or the VSMCs turning on the immune cell markers, CD68 and CD11b [16]. Increased infiltration of immune cells is suggested by an increased mRNA expression of MCP1 in thrombosed arteries (Figure 2). To investigate this, we isolated VSMCs from both control arteries and arteries with plaque, thrombosis, and stenosis and compared the VSMCs phenotype markers. Different phenotype markers of VSMCs showed changed expression compared to the expression of arterial tissue expression. Only S100A4 was found to have a similar expression. These findings support previously reported studies [9]. The differential expression of these markers in arterial tissue versus isolated cells might be due to VSMCs isolation or the different microenvironment in the in-vitro versus in-vivo studies. Since in-vitro phenotypic change occurs in stem cells during differentiation but not proliferation [33], the changed expression level in VSMCs phenotype markers found in this study may pertain to the microenvironment of immune cell infiltration and pro-inflammatory cytokine secretion, in-vivo. The differential expression of VSMC phenotypes in arterial tissues and isolated VSMCs are of significance when investigating the efficacy of a small molecule as a therapeutic drug because the effects of the small molecule under investigation on VSMCs phenotype may differ in-vivo.

The differential mRNA expression of VSMCs phenotypes markers in arterial tissues and isolated VSMCs suggested the role of immune cells and secreted cytokines [20,34]. Therefore, we investigated the role of IL-6, IL-1β, and TNF-α by treating isolated VSMCs with these cytokines. IL-6, IL-1β, and TNF-α showed different effects on the gene expression of VSMCs phenotype markers (Figure 4). The expression of contractile markers TAGLN and Myh11 were not significantly changed but there was a robust change in the mRNA expression of proliferative and de-differentiated VSMCs phenotype markers mainly for CK-18, CALM2, KLF4, and CD68. These findings suggest that the cytokines secreted from infiltrating immune cells after intimal injury influence the expression of VSMCs phenotype markers and may regulate phenotypic switch and plasticity. Since VSMC plasticity and phenotypic switch regulate vessel remodeling, targeting VSMC phenotype switch by modulating infiltrating immune cells infiltration may have therapeutic significance. The effect of individual cytokine on VSMC phenotype markers was different than when the VSMCs were co-cultured with M1 macrophages (Figure 5). This may be due to the cumulative effect of cytokines secreted by macrophages on VSMCs [18,35]. These findings suggest the importance of interaction between immune cells-VSMCs [10] and that targeting a single factor for a favorable outcome may not be effective and that we should have a multifactorial approach for outward vessel remodeling to induce AVF patency.

Since many of the genes expressed on VSMCs and the plasticity of VSMCs have transcriptional and epigenetic regulation [36,37] and these processes are involved in AVF maturation failure [22,23], we investigated the change in protein expression of VSMCs phenotype markers after treating VSMCs with IL-6, IL-1β, and TNF-α. Treatment of VSMCs with proinflammatory cytokines mimics the in-vivo environment for the presence of immune cells and cytokines secreted by these cells in the vicinity of AVF [26,27]. The results revealed the regulatory role of pro-inflammatory cytokines on the expression of VSMCs phenotype markers The effect on KLF4 protein expression was not significant, but this may be due to the treatment for only 24 h. However, increased KLF4 expression with IL-6 suggests that chronic exposure of VSMCs to IL-6 in vivo in AVF may cause VSMCs to acquire dedifferentiated phenotype secreting cytokines, in part, and precipitating chronic inflammation. These findings are supported by the dual immunopositivity of VSMCs for α-SMA with CD68, IL-6, IL-1β, and TNF-α.

To further investigate the regulatory effects of inflammatory cytokines on VSMC phenotype [20], we treated VSMCs with inhibitors of cytoplasmic kinases [38,39,40] and then treated them with IL-6, IL-1β, and TNF-α. Inhibition of cytoplasmic kinases PI3K (LY294002), JNK (SP600125), NF-κB (BAY 118075 and PDTC), and ERK (UO-126) with their inhibitors (Figure 7, Figure 8 and Figure 9) revealed attenuation of the effects of IL-6, IL-1β, and TNF-α on VSMCs phenotype markers suggesting that the effects of pro-inflammatory cytokines on VSMCs phenotypic change may be mitigated by blocking downstream signaling. Of note, the suppressive effects of inhibition were not the same for all genes and we noticed varying mitigating effects dependent on the type of kinase inhibition and cytokine stimulation (Figure 7, Figure 8 and Figure 9). The varying effects may be due to the involvement of cytoplasmic kinases in the downstream signaling of IL-6, IL-1β, and TNF-α. IL-6 downstream signaling through PI3K, MAPK (ERK), and NF-κB induces inflammation involving endothelial cells and precipitates abnormal coagulation and vascular leakage [41,42]. IL-1β, a therapeutic target in atherosclerosis and other cardiovascular diseases, signals through JNK and NF-κB and inhibition of IL-1β may decrease the production of MMP3, SMC chemotaxis, and inflammasome-mediated inflammation [43,44]. TNF-α downstream signaling through PI3K, JNK, and NF-κB regulates angiogenesis, cell proliferation, inflammatory response, cell survival, and apoptosis. These processes play a critical role in vessel remodeling and plaque progression [45,46]. Involvement of foam cells, macrophage-like-, synthetic-, endothelial cell-like angiogenic, and myofibroblasts VSMCs along with various inflammatory cytokines suggest the possible effects of these cytokines on VSMCs phenotype [47]. Further, differentiation of VSMCs involving various factor including low-density lipoprotein, KLF4, OCT4, TCF21, and other hormones indicate that not only inflammatory cytokines, but other factors also regulate VSMC de-differentiation or activation [47]. The role of other regulatory factors is supported by the fact that the effect of IL-6, IL-1β, and TNF-α on VSMCs phenotype markers was differentially mitigated by various inhibitors. Attenuated suppression by LY294002 (PI3K inhibitor), SP600125 (JNK inhibitor), BAY 118075 (NF-κB inhibitor), PDTC (NF-κB inhibitor), and UO-126 (ERK inhibitor) may also be due to the common involvement of PI3K, MAPK (ERK), JNK, and NF-κB in IL-6, IL-1β, and TNF-α downstream signaling.

These findings indicate that cytokines secreted from immune cells infiltrating the injury site may regulate VSMCs phenotype, and chronicity of inflammation causes an increased number of de-differentiated VSMCs altogether perpetuating persistent inflammation. Thus, targeting the immune cell infiltration or modulation for favorable immune cell infiltration should have therapeutic significance. The different effects on the gene expression of various markers with various cytokines in several doses also suggest context-dependent effects of pro-inflammatory cytokines. Inhibiting immune cell infiltration to attenuate cytokine production and atherosclerosis progression is supported by the reports that targeting inflammatory cytokines attenuates plaque progression [3,5,48,49]. Targeting inflammatory cytokines and inflammation to attenuate adverse remodeling and early thrombosis in AVF is supported by our recent findings on the role of inflammation in AVF failure [23]. The notion of targeting immune cell infiltration is supported by the recent findings on the beneficial use of resolvins, the pro-resolving mediators regulating immune cell infiltration and favoring the resolution phase in AVF [50,51,52].

The RT-qPCR findings, as discussed above, suggested the presence of varying VSMC phenotypes in arteries with plaques, thrombosis, and stenosis harvested from AVF junction and the role of pro-inflammatory cytokines in the plasticity and phenotype switch of VSMCs. To further support the hypothesis that pro-inflammatory cytokines and immune cells regulate VSMCs phenotypic switch, we dual-immunostained the carotid arteries with plaques and the co-localization of α-SMA with CD68 in arterial plaque (Figure 10). The results suggest that VSMCs positive for αSMA are also positive for CD68. These findings suggest the notion that de-differentiated VSMCs acquire monocyte/macrophage markers. Of note, all α-SMA positive VSMCs were not positive for CD68 and all CD68+ cells were not dual positive for α-SMA. These findings suggest the presence of three kinds of cells (αSMA+ VSMCs, CD68+ macrophages, and αSMA+CD68+ de-differentiated VSMCs) in the plaque area. Further, dual-immunopositivity of few cells for α-SMA with IL-6, IL-1β, and TNF-α (Figure 11) indicates the possibility of secretion of these cytokines from de-differentiated VSMCs, and this notion is supported by the RT-PCR findings of KLF4 expression on isolated VSMCs. However, these findings warrant in-depth investigation using in-vivo studies. Increased expression of cytokines in plaque areas may also be due to increased cytokine secretion from infiltrating macrophages and this notion is supported by the dual-immunopositivity for CD68 with IL-6, IL-1β, and TNF-α (Figure 12). The findings of dual immunofluorescence support the notion of an association and interaction between immune cells and VSMCs after intimal injury.

Overall, the findings of this study suggest the involvement of immune cells and pro-inflammatory cytokines in arterial plaque, thrombosis, and stenosis. Further, the results also suggest that pro-inflammatory cytokines regulate VSMCs phenotype switch during the AVF maturation process and inhibition of inflammatory cytokine downstream signaling mitigates the effects. Thus, targeting immune cell infiltration and phenotypic switch of VSMCs to decrease the number of de-differentiated VSMCs and chronicity of inflammation will have therapeutic significance in enhancing AVF maturation by attenuating early thrombosis and stenosis, and decreasing plaque progression. Additionally, these findings also support the notion that the VSMCs phenotype switch plays an important role in vessel remodeling, thrombosis, stenosis, and AVF maturation.

## 4. Materials and Methods

**Tissue collection and VSMCs isolation**: Femoral arteries with and without atherosclerotic plaque were harvested after euthanasia from Yucatan miniswine and microswine. Yucatan miniswine were purchased from Premier Bio-resources (Ramona, CA, USA) and Yucatan microswine were purchased from Sinclair Bio-resources, used for other ongoing studies in the lab. The miniswine were fed with Mini-Pig Grower Diet (Test Diet # 5801) and microswine were fed with high fat atherogenic diet (D17012601 purchased from Research Diets Inc., Ramona, CA, USA) and allowed to drink water *ad libitum*. All animal interventions were performed as per the National Institutes of Health and USDA for the care and use of experimental animal guidelines. The research protocol for the ongoing study was approved by the Institutional Animal Care and Use Committees (IACUC) at the Western University of Health Sciences (Protocols No. R20IACUC038 and R19IACUC026). Femoral and carotid arteries harvested after euthanasia (Appendix A) were collected in 3× DMEM (Dulbecco’s Modified Eagle Medium with 3% penicillin-streptomycin without fetal bovine serum) and transferred to the laboratory on ice. The collected arteries were washed two times with 3× DMEM. The adventitia was cleaned, and the media and interna were chopped to paste in 1 mL complete DMEM (DMEM with 10% fetal bovine serum and 1% penicillin-streptomycin) in a sterile culture dish. The arterial paste was transferred to a sterile 50 mL falcon tube and volume was made up to 5 mL followed by centrifugation at 1200 rpm for 10 min. After centrifugation, the supernatant was discarded and 20 mL of collagenase II (from *Clostridium histolyticum*, C0130 Sigma Aldrich) with a concentration of 1 mg/mL was added to the pellet. The pellet was gently dissolved, and the tube was incubated for 1 h at 37 °C with intermittent shaking every 15 min. This was followed by centrifugation at 1200 rpm for 10 min. The supernatant was discarded, and the pellet was mixed with complete SMCM (Smooth Muscle Cell Medium, Catalog #1101, ScienCell) with growth factors and fetal bovine serum. The cells were transferred to a T75 flask using a 100 μm filter and cells were grown to 70–805 confluency while changing media every 48 h. The cells were trypsinized and 50% VSMCs from the primary cell culture were frozen for future use and the remaining cells in passages 3–4 were used for all experiments. The phenotype of cells was determined by staining with alpha-smooth muscle actin (α-SMA, ab7817). VSMCs were also checked for morphology and only a few fibroblasts/myofibroblasts were seen in primary culture but the number decreased while SMCM was used in successive passages.

**Peripheral blood mononuclear cells (PBMCs) isolation**: PBMCs from the blood collected from microswine before euthanasia was isolated using the density gradient method. The collected whole blood with insulin was diluted with phosphate-buffered saline (PBS) and gently layered over an equal volume of Ficoll (Cytiva #17144002) in a 50 mL Falcon tube. This was followed by centrifugation at 500× *g* for 40 min. The second layer containing PBMCs on the top was collected and grown in complete DMEM for 5–7 days in a humidified incubator at 37 °C with 5% CO_2_. The cells adherent to the plate were collected and 10,000 cells were plated in a transwell chamber with complete DMEM to convert monocytes to macrophages.

**PBMC to M1 macrophage differentiation**: The PBMCs plated on the upper chamber of the transwell were treated first with 80 ng/mL M-CSF (BioRad PPP031, Hercules, CA, USA) for four days followed by 100 ng/mL LPS (Sigma L5293-2ML, Burlington, MA, USA) and 20 ng/mL IFN-γ (R & D system, 985-PI-050/CF, MN, USA) for 4–6 days changing medium every 48 h [53]. The differentiation of monocytes to macrophages was observed microscopically for the change in morphology. Once the cells acquired M1 morphology, the cells were co-cultured with VSMCs.

**Coculture experiments**: Coculture in vitro experiments were conducted to investigate the effect of M1 macrophage on VSMC phenotype. The M1 macrophages on the top layer of the transwell were co-cultured with VSMCs in the lower chamber in a six-well plate with 1 × 10^5^ cells per well. The cells were incubated with appropriate control without any macrophage co-culture from the same passage in a separate culture dish to compare with the VSMCs in co-culture. The cells were incubated for 24 h using a serum-free DMEM medium. After 24 h, the supernatant was collected and stored at −80 °C. The VSMCs were washed once with sterile PBS and the cells were collected using 1 mL TRIzol for RNA isolation.

**Stimulation and inhibition studies**: Since immune cells recruited after intimal injury secrete various inflammatory cytokines, the effect of the cytokines was examined by treating the cells with IL-6, TNF-α, and IL-1β. For the stimulation experiment, VSMCs were grown in a T75 flask and once the cells reached 80–90% confluence, cells were trypsinized and 1 × 10^6^ cells were plated in each 6-well plate. After growing cells overnight cells were treated with IL-6, TNF-α, and IL-1β in three doses (1 ng/mL, 10 ng/mL, and 100 ng/mL) using a serum-free DMEM medium. VSMCs were incubated for 24 h and then cells were collected in TRIzol for total RNA isolation. For inhibition studies, cells plated in 6 well plates were treated with inhibitors of cytoplasmic kinases involved in the downstream signaling of IL-6, TNF-α, and IL-1β including LY294002 (PI3K inhibitor, 1.4 μM), SP600125 (JNK inhibitor, 40 nM), BAY 118075 (NF-κ B inhibitor, 10 μM), PDTC (NF-κ B inhibitor, 10 μM), and UO-126 (ERK inhibitor, 70 μM). The doses of these compounds were determined based on the EC_50_ values and our previous studies [5]. For inhibition studies, the cells were first treated with these compounds for 2 h and then with IL-6 (10 ng/mL), TNF-α (100 ng/mL), and IL-1β (100 ng/mL). The doses of IL-6, TNF-α, and IL-1β for inhibition studies were based on the stimulation studies and the dose with the best effect was chosen to do inhibitory studies (please see the result section and Appendix A). After incubating cells for 24 h, cells were collected in TRIzol for total RNA isolation.

**RNA isolation and cDNA preparation**: RNA was isolated from both VSMCs and arterial tissues collected from miniswine. The tissues were homogenized using a homogenizer with tissues chopped and kept in 500 μL TRIzol in a 1.5 mL Eppendorf tube. Once the tissues were homogenized, an additional 500 μL TRIzol was added to make it 1 mL. In the case of VSMCs, the cells were collected in 1 mL TRIzol. After this both VSMCs and tissues were processed similarly. The cells and tissues were incubated in TRIzol for 10–15 min at room temperature. Following this, 200 μL of chloroform was added, mixed vigorously, incubated for 10 min at room temperature, and then centrifuged at 12,000 rpm for 15 min at 4 °C. The upper transparent layer was collected, mixed with 500 μL isopropanol, mixed gently by inverting tube 5–6 times, incubated for 5 min at room temperature, and then centrifuged at 12,000 rpm for 10 min. The supernatant was discarded, and the pellet was washed with 70% ethanol, centrifuged at 12,000 rpm for 10 min followed by air drying the pellet after removal of the ethanol. The pellet was mixed with 10–20 μL RNA-DBA free water and the yield of RNA was estimated using NanoDrop 2000 (Thermo Scientific). 2 μg cDNA was synthesized using the iScript cDNA synthesis kit (BioRad #1708891) following the manufacturer’s instructions.

**Real-Time quantitative Polymerase Chain Reaction** (qRT-PCR): qRT-PCR for the mRNA expression of various genes was carried out using AzuraView Green Fast qPCR Blue Mix (AZ-2405) using a Real-time PCR system (CFX96, BioRad Laboratories, Hercules, CA, USA). qRT-PCR was conducted with the cycling of 5 min at 95 °C for initial denaturation, 40 cycles of 30 s at 95 °C, 30 s at 55–60 °C, and 30 s at 72 °C followed by melting curve analysis in triplicates for each gene. The primers were designed using NCBI (https://www.ncbi.nlm.nih.gov/ assessed on 3 May 2022) and ordered from Integrated DNA Technology (IDT Coralville, Iowa 52241 USA). The nucleotide sequence for the primers used ate listed in Appendix A. The fold change in mRNA expression was calculated using 2^−^^CT^ method after normalizing the CT values with housekeeping gene 18S.

**Tissue processing**: The tissue samples including femoral artery proximal (PFA) and distal (DFA) to the anastomosis, femoral vein proximal to anastomosis (PFV), and contralateral femoral artery and femoral vein collected after pig euthanasia were fixed in 4% formalin and processed in a tissue processor. The tissues were treated through multiple changes in ethanol (70%, 90%, and 100%) for dehydration and by multiple changes in xylene for clearing. Tissue blocks were made from the processed tissues using paraffin wax. The paraffin wax blocks with tissues were sectioned at 5 µm using a tungsten carbide knife (LeicaTM, Germany) in a Leica RM2265 rotary microtome (LeicaTM, Germany) and attached to slides. The tissue sections were heat blocked for 60 min and used for histology and immunostaining.

**Immunofluorescence studies**: Immunofluorescence studies were conducted on the carotid arteries collected from Yucatan microswine on hypercholesterolemic diet and with intimal injury being used for another ongoing study in the lab (IACUC approval # R19IACUC026). Tissue section of the arteries from intimal injury site with plaque formation and contralateral carotids were used from 7 swine. For immunofluorescence, paraffin sections were deparaffinized, dehydrated, and antigen retrieved. After antigen retrieval, tissue sections were washed with phosphate-buffered saline (PBS), blocked with a 10% blocking agent for 1 h, and then incubated with primary antibodies (Appendix A) overnight at 4 °C. Following this, the sections were washed three times five minutes each with PBS and then incubated with secondary antibodies Alexa Fluor 596 (red) and Alexa Fluor 488 (green) at a dilution of 1:1000 for 1 h at room temperature. The slides were washed with PBS two times for five minutes each with mild shaking. The slides were mounted with VECTASHIELD PLUS Antifade Mounting Medium with DAPI (H-2000). 4,6-diamidino-2-phenylindole (DAPI) stained the nucleus blue. The sections were covered with a cover slip and scanned using Leica inverted microscope. Negative controls for each primary and secondary antibody were stained following the standard procedure by omitting secondary antibodies and using primary antibodies only for all proteins and negative controls for Alexa Fluor were run using only secondary antibodies while omitting primary antibodies (Appendix A).

**Western Blot analysis**: VSMCs were plated in a 100 mm culture dish and cultured till 80% confluence. The cells were treated with pro-inflammatory cytokines IL-6 (10 ng/mL), TNF-α (100 ng/mL), and IL-1β (100 ng/mL) for 24 h with appropriate control. For protein extraction, the media was removed, cells were washed with ice-cold 1× PBS, and cells were lysed using 1x RIPS buffer. The cells with RIPA buffer were centrifuged for 15 min at @12,000 rpm and the supernatant was collected. After estimating the protein using Bradford assay (BioRad # 5000006, Hercules, CA, USA), 30 μg protein was loaded and ran on 4–15% Mini-PROTEAN TGX Precast Protein Gels, (BioRad #4561084). The protein was transferred to the polyvinylidene difluoride (PVDF) membrane and then the membrane was blocked with 5% milk for 1 h. This was followed by incubation with primary antibodies (Appendix A) overnight at 4 °C and then washing with Tris-buffered saline with 0.1% Tween 20 (TBST) and incubation with secondary antibodies. ECL was used to develop the chemiluminescence and imaging was performed using GelDoc (BioRad# 1708195).

**Statistical analysis**: Data are presented as the mean ± SD. Data were analyzed using GraphPad Prism 9. The comparison between two groups for the expression was performed using One-way ANOVA with Bonferroni’s post hoc correction and students’ *t*-test was used for statistical significance. A probability (p) value of < 0.05 was accepted as statistically significant. * *p* < 0.05, ** *p* < 0.01, *** *p* < 0.001, and **** *p* < 0.0001.

## 5. Conclusions

The findings of this study highlight the notion that the phenotypic switch in VSMCs to a dedifferentiated phenotype mimicking the function of immune cells is mediated by cytokines secreted by infiltrating immune cells. The presence of immune cells in AVF tissues and the presence of early thrombosis and stenosis of AVF vessels indicate the need for focusing therapeutic strategies targeting immune cell infiltration and VSMCs phenotype switch to enhance maturation in AVF.

## Figures and Tables

**Figure 1 ijms-23-12012-f001:**
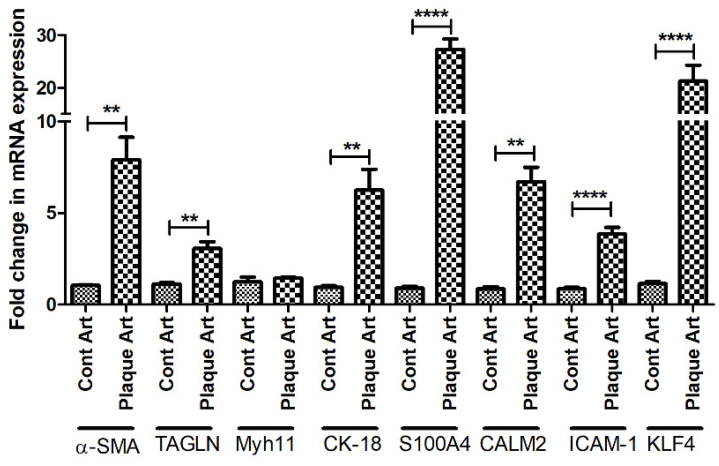
RT-qPCR data for VSMCs phenotype markers in arteries with and without atherosclerotic plaques. Alpha-smooth muscle actin (α-SMA), transgelin (TAGLN), Myosin Heavy Chain 11 smooth muscle (Myh11), cytokeratin (CK)-18, S100 protein A4 (S100A4), calmodulin 2 (CALM2), Intercellular Adhesion Molecule (ICAM)1, Krüppel-like factor 4 (KLF4). α-SMA, TAGLN, and Myh11 (markers for contractile vascular smooth muscle cells (VSMCs)); CK-18, S100A4, CALM2, and ICAM1 (markers for proliferative VSMCs); and KLF4 (a marker for de-differentiated VSMCs). All data are presented as mean ± standard deviation (SD). *p* < 0.05 is considered statistically significant. ** *p* < 0.01 and **** *p* < 0.0001.

**Figure 2 ijms-23-12012-f002:**
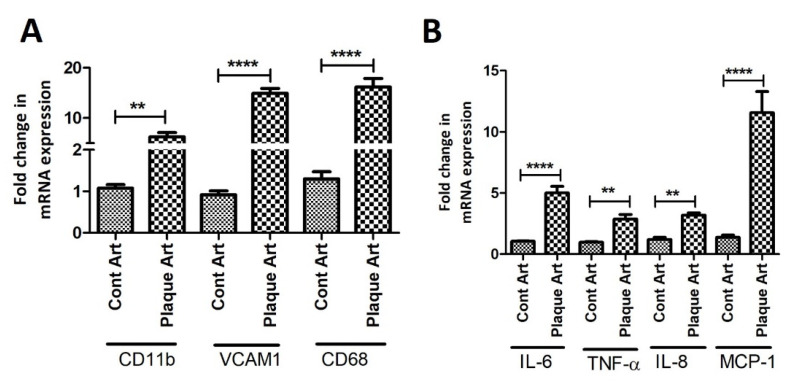
RT-qPCR data for VSMCs phenotype and cytokines in arteries with and without plaque. Panel (**A**) shows qRT-PCR data for CD11b, VCAM1, and CD68, and Panel (**B**) shows qRT-PCR data for cytokines IL-6, IL-8, TNF-α, and MCP1. Vascular Cell Adhesion Molecule (VCAM)1, interleukin (IL), tumor necrosis factor (TNF)-α, CD11b- maker for monocytes, CD68-marker for macrophages, MCP1-monocyte chemoattractant protein 1. All data are presented as mean ± standard deviation (SD). *p* < 0.05 is considered statistically significant. ** *p* < 0.01 and **** *p* < 0.0001.

**Figure 3 ijms-23-12012-f003:**
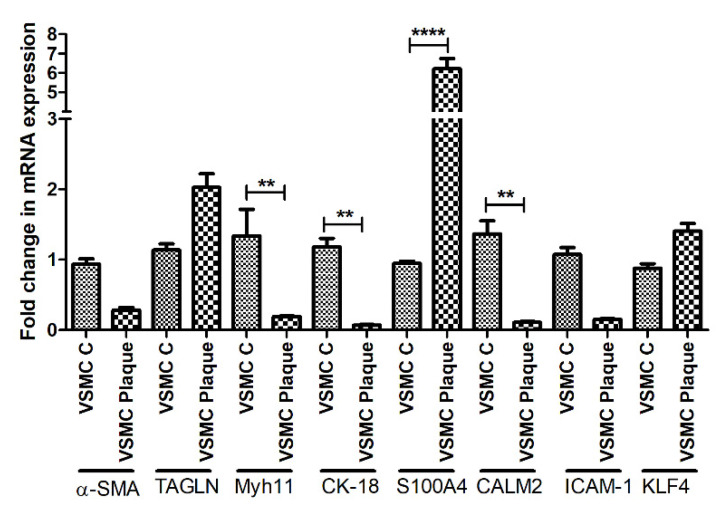
RT-qPCR data for VSMCs phenotype markers in VSMCs isolated from control and arteries with plaque. Alpha-smooth muscle actin (α-SMA), transgelin (TAGLN), Myosin Heavy Chain 11 smooth muscle (Myh11), cytokeratin (CK)-18, S100 protein A4 (S100A4), calmodulin 2 (CALM2), Intercellular Adhesion Molecule (ICAM)1, Krüppel-like factor 4 (KLF4). α-SMA, TAGLN, and Myh11 (markers for contractile vascular smooth muscle cells (VSMCs)); CK-18, S100A4, CALM2, and ICAM1 (markers for proliferative VSMCs); and KLF4 (a marker for de-differentiated VSMCs). All data are presented as mean ± standard deviation (SD). *p* < 0.05 is considered statistically significant. ** *p* < 0.01 and **** *p* < 0.0001.

**Figure 4 ijms-23-12012-f004:**
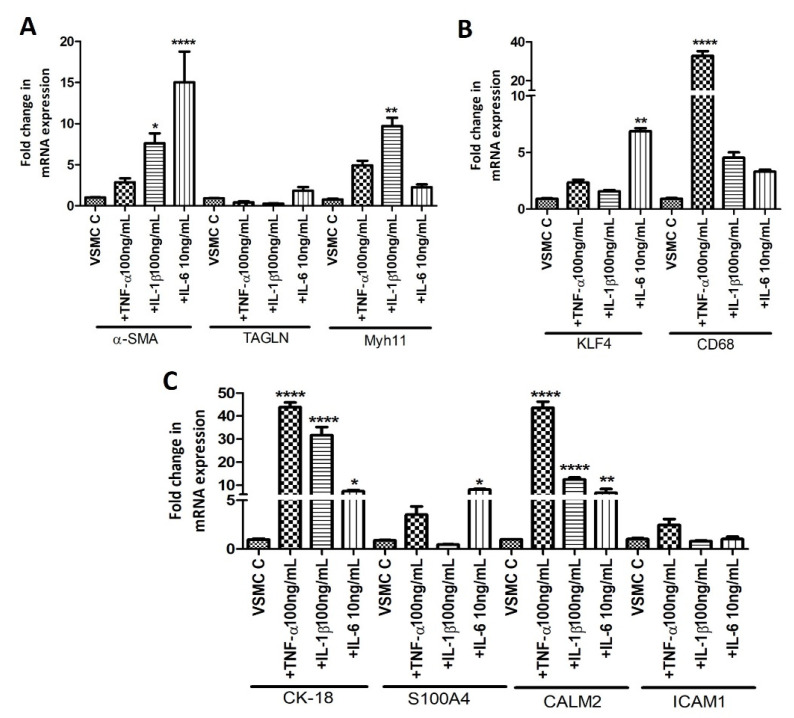
RT-qPCR data for the effects of pro-inflammatory cytokines IL-6, IL-1β, and TNF-α on VSMCs phenotype markers. Panel (**A**) shows qRT-PCR data for α-SMA, TAGLN, and Myh11; Panel (**B**) shows qRT-PCR for CK-18, S100A4, CALM2, and ICAM1; and Panel (**C**) shows qRT-PCR for KLF4 and CD68. Alpha-smooth muscle actin (α-SMA), transgelin (TAGLN), Myosin Heavy Chain 11 smooth muscle (Myh11), cytokeratin (CK)-18, S100 protein A4 (S100A4), calmodulin 2 (CALM2), Intercellular Adhesion Molecule (ICAM)1, Krüppel-like factor 4 (KLF4). α-SMA, TAGLN, and Myh11 (markers for contractile vascular smooth muscle cells (VSMCs)); CK-18, S100A4, CALM2, and ICAM1 (markers for proliferative VSMCs); and KLF4 (a marker for de-differentiated VSMCs). All data are presented as mean ± standard deviation (SD). *p* < 0.05 is considered statistically significant. * *p* < 0.05, ** *p* < 0.01 and **** *p* < 0.0001.

**Figure 5 ijms-23-12012-f005:**
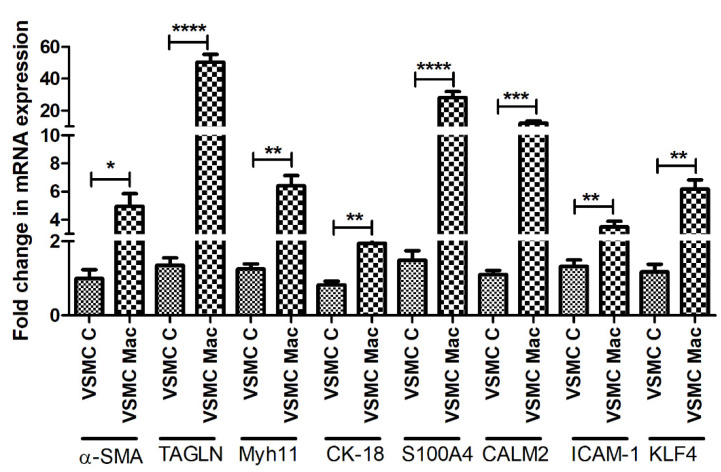
RT-PCR analysis of the effect of M1 macrophages on VSMCs phenotype markers. Alpha-smooth muscle actin (α-SMA), transgelin (TAGLN), Myosin Heavy Chain 11 smooth muscle (Myh11), cytokeratin (CK)-18, S100 protein A4 (S100A4), calmodulin 2 (CALM2), Intercellular Adhesion Molecule (ICAM)1, Krüppel-like factor 4 (KLF4). α-SMA, TAGLN, and Myh11 (markers for contractile vascular smooth muscle cells (VSMCs)); CK-18, S100A4, CALM2, and ICAM1 (markers for proliferative VSMCs); and KLF4 (a marker for de-differentiated VSMCs). All data are presented as mean ± standard deviation (SD). *p* < 0.05 is considered statistically significant. * *p* < 0.05, ** *p* < 0.01, *** *p* < 0.001, and **** *p* < 0.0001.

**Figure 6 ijms-23-12012-f006:**
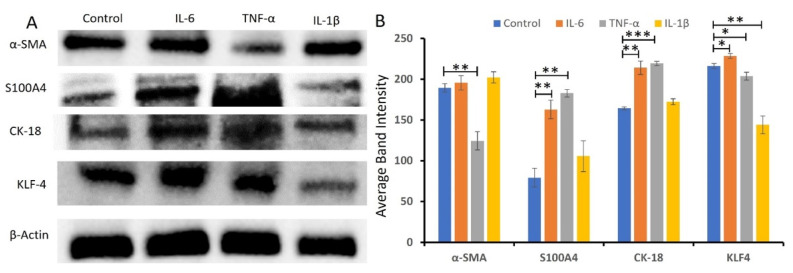
Western blot analyses of the effects of inflammatory cytokines on VSMCs phenotype markers. Panel (**A**)—Western blot images of VSMC markers; Panel (**B**)—percent changes in protein expression compared to control. Control protein was considered as 100% for the analysis. Alpha-smooth muscle actin (α-SMA), cytokeratin (CK)-18, S100 protein A4 (S100A4), Krüppel-like factor 4 (KLF4), interleukin (IL), and tumor necrosis factor (TNF)-α. * *p* < 0.05, ** *p* < 0.01, *** *p* < 0.001.

**Figure 7 ijms-23-12012-f007:**
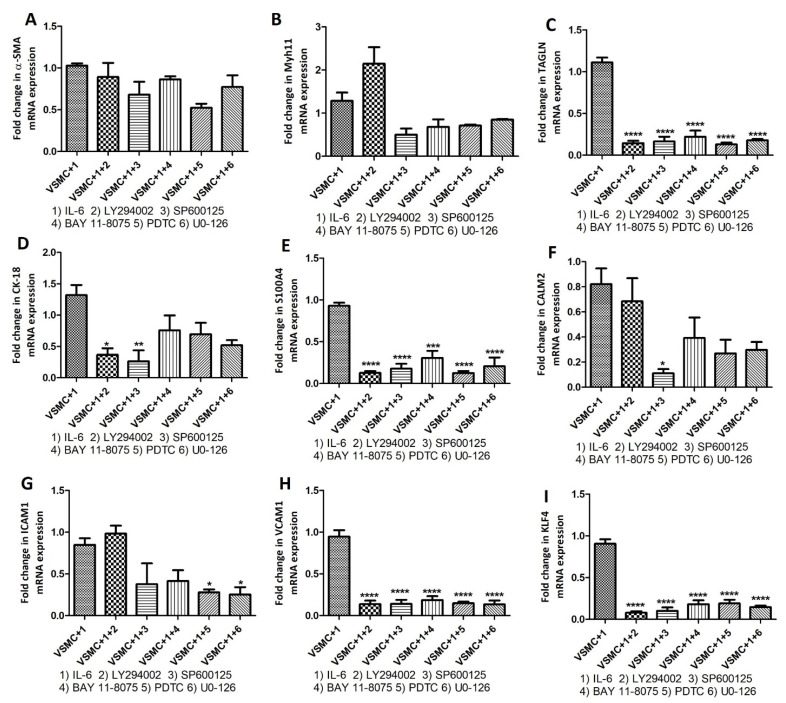
RT-qPCR to investigate the effect of IL-6 on VSMCs phenotype markers after inhibiting cytoplasmic kinases. RT-PCR data for alpha-smooth muscle actin (α-SMA; panel **A**), Myosin Heavy Chain 11 smooth muscle (Myh11; panel **B**), transgelin (TAGLN; panel **C**), cytokeratin 18 (CK-18; panel **D**), S100 protein A4 (S100A4; panel **E**), calmodulin 2 (CALM2; panel **F**), Intercellular Adhesion Molecule 1 (ICAM1; panel **G**), Vascular Cell Adhesion Molecule 1 (VCAM1; panel **H**), Krüppel-like factor 4 (KLF4; panel **I**). α-SMA, TAGLN, and Myh11 (markers for contractile vascular smooth muscle cells (VSMCs)); CK-18, S100A4, CALM2, and ICAM1 (markers for proliferative VSMCs); and KLF4 (marker for de-differentiated VSMCs). Interleukin (IL)-6, LY294002 (PI3K inhibitor), SP600125 (JNK inhibitor), BAY 118075 (NF-κB inhibitor), PDTC (NF-κB inhibitor), and UO-126 (ERK inhibitor). All data are presented as mean ± standard deviation (SD). *p* < 0.05 is considered statistically significant. * *p* < 0.05, ** *p* < 0.01, *** *p* < 0.001, and **** *p* < 0.0001.

**Figure 8 ijms-23-12012-f008:**
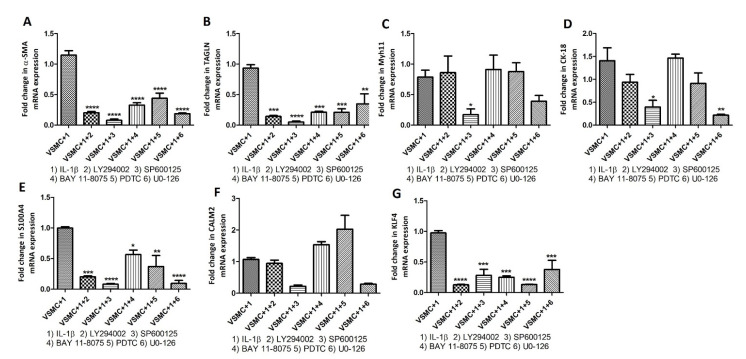
RT-qPCR to investigate the effect of IL-1β on VSMCs phenotype markers after inhibiting cytoplasmic kinases. RT-PCR data for alpha-smooth muscle actin (α-SMA; panel **A**), transgelin (TAGLN; panel **B**), Myosin Heavy Chain 11 smooth muscle (Myh11; panel **C**), cytokeratin 18 (CK-18; panel **D**), S100 protein A4 (S100A4; panel **E**), calmodulin 2 (CALM2; panel **F**), and Krüppel-like factor 4 (KLF4; panel **G**). α-SMA, TAGLN, and Myh11 (markers for contractile vascular smooth muscle cells (VSMCs)); CK-18, S100A4, and CALM2 (markers for proliferative VSMCs); and KLF4 (marker for de-differentiated VSMCs). Interleukin (IL)-1β, LY294002 (PI3K inhibitor), SP600125 (JNK inhibitor), BAY 118075 (NF-κB inhibitor), PDTC (NF-κB inhibitor), and UO-126 (ERK inhibitor). All data are presented as mean ± standard deviation (SD). *p* < 0.05 is considered statistically significant. * *p* < 0.05, ** *p* < 0.01, *** *p* < 0.001, and **** *p* < 0.0001.

**Figure 9 ijms-23-12012-f009:**
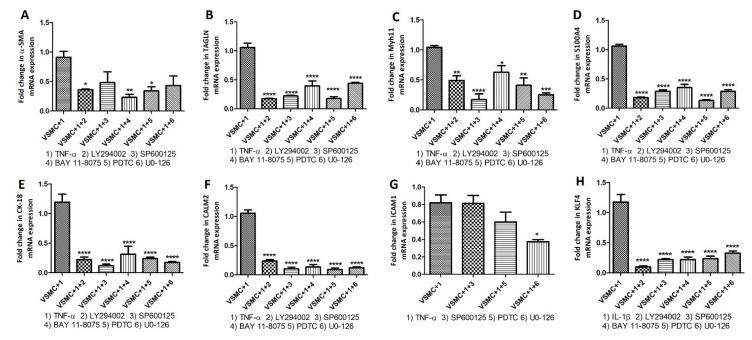
RT-qPCR to investigate the effect of TNF-α on VSMCs phenotype markers after inhibiting cytoplasmic kinases. RT-PCR data for alpha-smooth muscle actin (α-SMA; panel **A**), transgelin (TAGLN; panel **B**), Myosin Heavy Chain 11 smooth muscle (Myh11; panel **C**), S100 protein A4 (S100A4; panel **D**), cytokeratin 18 (CK-18; panel **E**), calmodulin 2 (CALM2; panel **F**), Intercellular Adhesion Molecule 1 (ICAM1; panel **G**), Krüppel-like factor 4 (KLF4; panel **H**). α-SMA, TAGLN, and Myh11 (markers for contractile vascular smooth muscle cells (VSMCs)); CK-18, S100A4, CALM2, and ICAM1 (markers for proliferative VSMCs); and KLF4 (marker for de-differentiated VSMCs). Tumor necrosis factor (TNF)-α, LY294002 (PI3K inhibitor), SP600125 (JNK inhibitor), BAY 118075 (NF-κB inhibitor), PDTC (NF-κB inhibitor), and UO-126 (ERK inhibitor). All data are presented as mean ± standard deviation (SD). *p* < 0.05 is considered statistically significant. * *p* < 0.05, ** *p* < 0.01, *** *p* < 0.001, and **** *p* < 0.0001.

**Figure 10 ijms-23-12012-f010:**
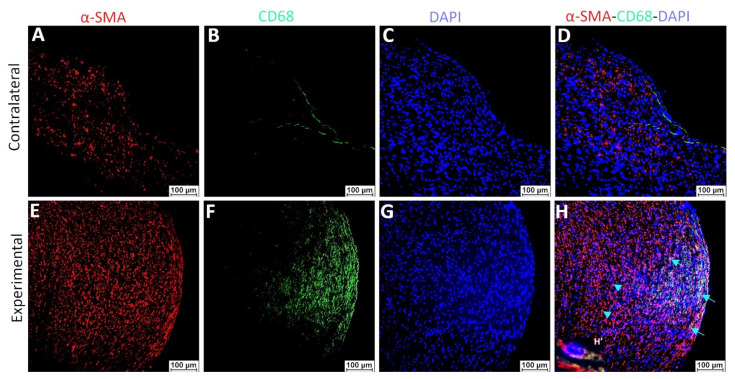
Dual immunofluorescence staining of α-SMA with CD68. Immunofluorescence for alpha-smooth muscle actin (α-SMA, panels **A**,**E**), CD68 (a marker for macrophages, panels **B**,**F**), DAPI (panels **C**,**G**), and merged images (panels **D**,**H**). Control- arteries without plaque (Contralateral) and experimental- arteries with plaque (Experimental). DAPI (4′,6-diamidino-2-phenylindole). The arrows indicate dual positive cells while the arrowhead indicates immunopositivity for one protein. Inset in panel H (H’) shows the co-localization of α-SMA and CD68; the inset correspondence to the dual positive cells pointed by the right lower arrow).

**Figure 11 ijms-23-12012-f011:**
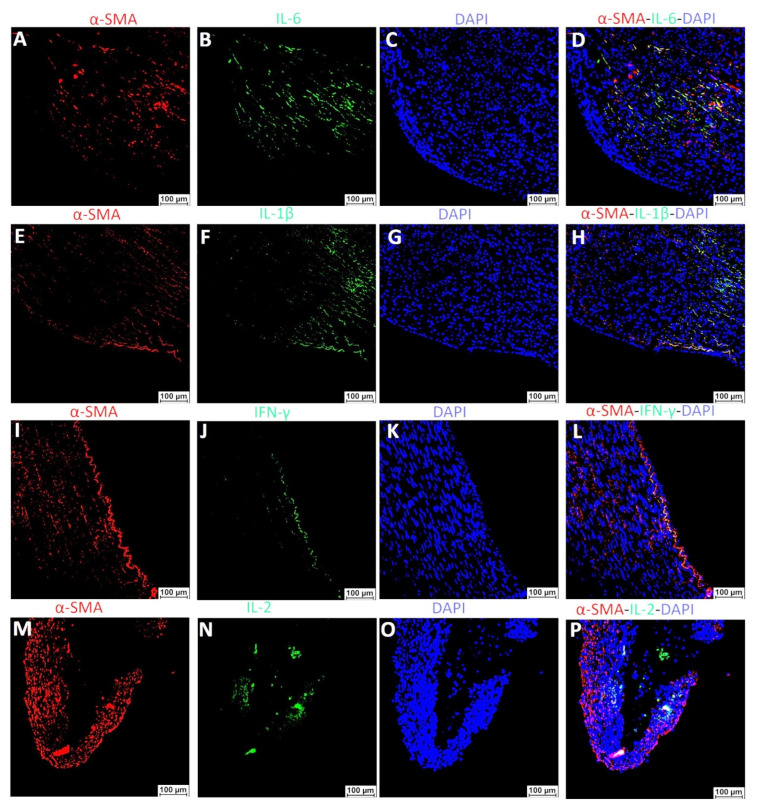
Dual-immunofluorescence staining of arteries with a plaque for α-SMA and pro-inflammatory cytokines IL-6, IL-1β, TNF-α, and IL-2. Immunofluorescence for alpha-smooth muscle actin (α-SMA, panels **A**,**E**,**I**,**M**), interleukin 6 (IL-6, panel **B**), interleukin 1β (IL-1β, panel **F**), tumor necrosis factor-alpha (TNF-α, panel **J**), interleukin 2 (IL-2, panel **N**), 4′,6-diamidino-2-phenylindole (DAPI, panels **C**,**G**,**K**,**O**), and merged images (panels **D**,**H**,**L**,**P**). The arrows indicate dual positive cells while the arrowhead indicates immunopositivity for one protein.

**Figure 12 ijms-23-12012-f012:**
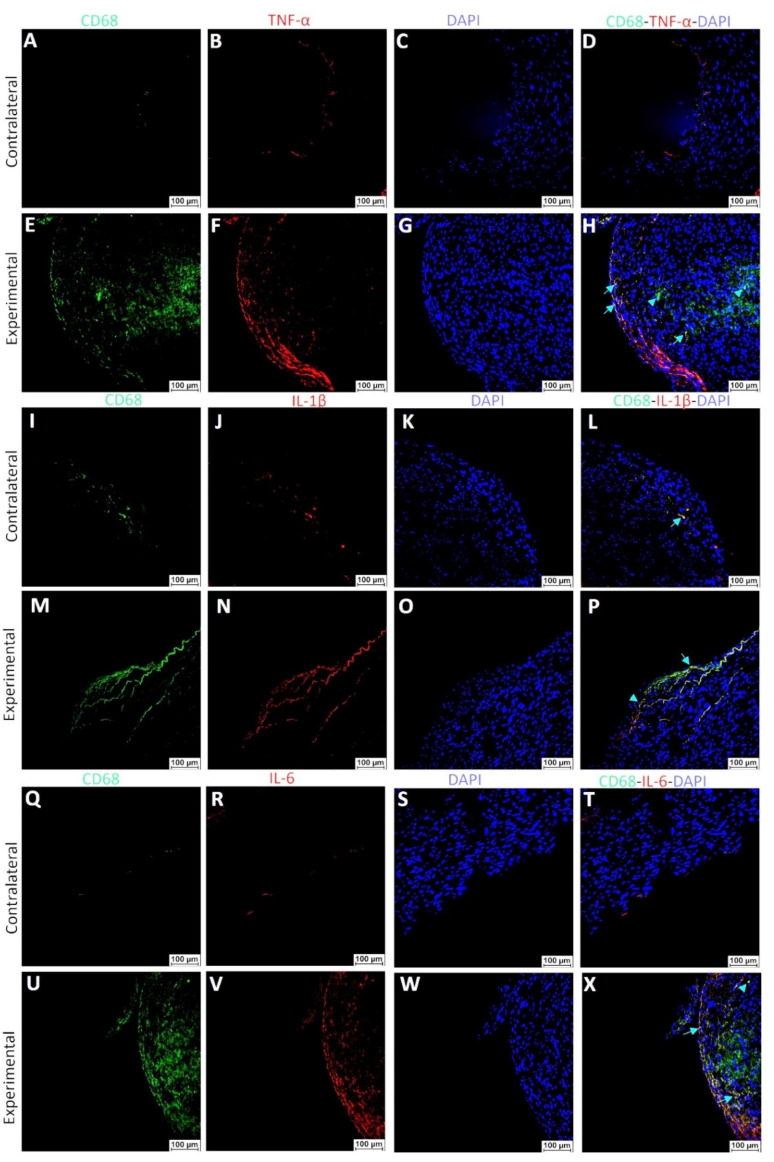
Dual-immunofluorescence of control arteries (contralateral) and arteries with plaque (experimental) for CD68 and IL-6, IL-1β, and TNF-α. Immunofluorescence for macrophage marker CD68 (panels **A**,**E**,**I**,**M**,**Q**,**U**), tumor necrosis factor-alpha (TNF-α, panels **B**,**F**), interleukin 1 beta (IL-1β, panels **J**,**N**), interleukin 6 (IL-6, panels **R**,**V**), 4′,6-diamidino-2-phenylindole (DAPI, panels **C**,**G**,**K**,**O**,**S**,**W**) and merged images (panels **D**,**H**,**L**,**P**,**T**,**X**). Control—arteries without plaque and experimental—arteries with plaque. The arrows indicate dual positive cells while the arrowhead indicates immunopositivity for one protein.

## Data Availability

All data related to this study have been included in this manuscript and the Appendix A.

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
