# Peer review of "Targeting the Crosstalk of Immune Response and Vascular Smooth Muscle Cells Phenotype Switch for Arteriovenous Fistula Maturation"

_ijms, 2022, doi:10.3390/ijms231912012_

Round 1

Reviewer 1 Report

The manuscript of Rai, Singh and Agrawal describes the identification and description of vascular smooth muscle cells that have been altered (activated or de-differentiated) by inflammatory conditions in vivo and in vitro, in the context of arteriovenous fistula (AVF) maturation. This has great relevance to atherothrombosis and stroke formation. The authors have done a very good job of explaining the rationale for the work. Furthermore, the experimental design is excellent with important contributions to the field. The conclusions are well supported. Overall, this is an excellent manuscript. I have a few comments.   1. The co-culture experiments using the transwell were well-designed. 2. Figure 3: First, I would suggest making the control bar the same pattern/colour for all genes and make the plaque bar the same pattern/colour for all genes. All the multiple colourings and patterns are confusing and don't allow the reader to identify the couplets for each gene (control vs. plaque). This could also be applied to Figure 1 and 2, but the pattterns are not confusing in those figures. Alignment of the tags on the bottom of Figure 3 for each sets of conditions is confusing. 3. Figure 6: Is the quantification of the bands from the western from just one experiment and blot? Multiple blots from different experiments must be used and then can be normalized for statistical comparison. Otherwise, the results are weak and can not really be informative. 4. Figures 7-9: very interesting and informative 5. Figure 10: Images are good, but an inset must be included to show a blow up (zoom in) of the colocalization. 6. The authors should detail specific results from Figures 7-9 in the discussion (ie. what inhibitor inhibits which cytokine signaling).

Author Response

Response to Reviewers’ Comments

Reviewer #1:

Overall Comment: The manuscript of Rai, Singh and Agrawal describes the identification and description of vascular smooth muscle cells that have been altered (activated or de-differentiated) by inflammatory conditions in vivo and in vitro, in the context of arteriovenous fistula (AVF) maturation. This has great relevance to atherothrombosis and stroke formation. The authors have done a very good job of explaining the rationale for the work. Furthermore, the experimental design is excellent with important contributions to the field. The conclusions are well supported. Overall, this is an excellent manuscript. I have a few comments.  

Response: Thank you for your encouraging comments and suggestions.

Comment 1. The co-culture experiments using the transwell were well-designed.

Response: Thank you for your kind comment.

Comment 2. Figure 3: First, I would suggest making the control bar the same pattern/colour for all genes and make the plaque bar the same pattern/colour for all genes. All the multiple colourings and patterns are confusing and don't allow the reader to identify the couplets for each gene (control vs. plaque). This could also be applied to Figure 1 and 2, but the pattterns are not confusing in those figures. Alignment of the tags on the bottom of Figure 3 for each sets of conditions is confusing.

Response: Thank you for your comments and suggestion. We have modified all figures. In the revised figures, control and treatment groups have similar pattern, as suggested.

Comment 3. Figure 6: Is the quantification of the bands from the western from just one experiment and blot? Multiple blots from different experiments must be used and then can be normalized for statistical comparison. Otherwise, the results are weak and cannot really be informative.

Response: Thank you for your suggestion. We have revised the figure and the results from three biological replicates have been used for calculation. The quantitative data from 3 separate experiments is provided.

Comment 4. Figures 7-9: very interesting and informative

Response: Thank you for your comment.

Comment 5. Figure 10: Images are good, but an inset must be included to show a blow up (zoom in) of the colocalization.

Response: Thank you for your comment and suggestion. We have inserted the inset showing the colocalization of α-SMA and CD68 (Figure 10 panel H-inset H’) in the revised manuscript.

Comment 6. The authors should detail specific results from Figures 7-9 in the discussion (ie. what inhibitor inhibits which cytokine signaling).

Response: Thank you for your comment and suggestion. We have included the text on the involvement of IL-6, IL-1β, and TNF-α, the downstream signaling involving cytoplasmic kinases PI3K, MAPK (ERK), JNK, and NF-κB, and the inhibitor specific for each kinase in the revised manuscript (Lines 363-383).

Reviewer 2 Report

Comments and suggestions

This study investigates the effects of the impaired immune response on Vascular Smooth Muscle Cells (VSMCs) isolated from femoral arteries, with or without atherosclerotic plaques. The authors showed the role of IL-6, IL-1 beta and TNF-alpha (as pro-inflammatory cytokines), as well as macrophages (immune cells secreting the above-mentioned cytokines) in VSMCs phenotype changes. They demonstrated that the inhibition of cytokine downstream signaling controls this change. This is not surprising, since the role of inflammation in atherosclerosis and cardiovascular diseases is widely accepted. Overall, the paper is interesting. However, some questions have to be addressed:     

  1. Why did the authors focus their attention on IL-6, IL-1 beta and TNF-alpha, excluding other relevant cytokines involved in the pathogenesis of atherosclerosis? You should emphasize the role of these cytokines in atherosclerosis and cardiovascular diseases. 
  2. It could be interesting to investigate the expression of the monocyte-chemotactic protein (MCP)-1 which lead to monocyte recruitment.
  3. In Figure 6, panel B, is each protein normalized to Beta-Actin and then to the control group?
  4. There are some grammatical errors that should be carefully edited.

Author Response

Reviewer #2: 

Overall Comment: This study investigates the effects of the impaired immune response on Vascular Smooth Muscle Cells (VSMCs) isolated from femoral arteries, with or without atherosclerotic plaques. The authors showed the role of IL-6, IL-1 beta and TNF-alpha (as pro-inflammatory cytokines), as well as macrophages (immune cells secreting the above-mentioned cytokines) in VSMCs phenotype changes. They demonstrated that the inhibition of cytokine downstream signaling controls this change. This is not surprising, since the role of inflammation in atherosclerosis and cardiovascular diseases is widely accepted. Overall, the paper is interesting. However, some questions have to be addressed:     

Response: Thank you for your comments and suggestions.

Comment 1: Why did the authors focus their attention on IL-6, IL-1 beta and TNF-alpha, excluding other relevant cytokines involved in the pathogenesis of atherosclerosis? You should emphasize the role of these cytokines in atherosclerosis and cardiovascular diseases. 

Response: Thank you for your comment and suggestion. We have included the text on the role of macrophages and proinflammatory cytokines in atherosclerosis and early AVF failure along with the rationale of assessing the effects of IL-6, IL-1β and TNF-α on vascular smooth muscle cell phenotypes.

Comment 2: It could be interesting to investigate the expression of the monocyte-chemotactic protein (MCP)-1 which lead to monocyte recruitment.

Response: Thank you for your comment and suggestion. The results for gene expression of MCP1 have been included in Figure 2 panel B in the revised manuscript.

Comment 3: In Figure 6, panel B, is each protein normalized to Beta-Actin and then to the control group?

Response: Thank you for your suggestion. We have revised the figure and the results from three biological replicates after normalizing to β-actin have been used.

Comment 4: There are some grammatical errors that should be carefully edited.

Response: Thank you for the comment. We have thoroughly checked the manuscript for the English language and grammar.

We hope that the revised manuscript is now suitable for publication in IJMS Section: Molecular Endocrinology and Metabolism Special Issue: Cellular and Molecular Mechanisms of Cardiovascular and Metabolic Diseases.